# A new conceptual framework for the transformation of groundwater dissolved organic matter

Liza K. McDonough [1,2✉], Martin S. Andersen[2,3], Megan I. Behnke [4], Helen Rutlidge[2,3], Phetdala Oudone[2,5], Karina Meredith[1], Denis M. O'Carroll [2,3], Isaac R. Santos[6], Christopher E. Marjo [7], Robert G. M. Spencer[4], Amy M. McKenna [8] & Andy Baker [2,5]

Groundwater comprises 95% of the liquid fresh water on Earth and contains a diverse mix of dissolved organic matter (DOM) molecules which play a significant role in the global carbon cycle. Currently, the storage times and degradation pathways of groundwater DOM are unclear, preventing an accurate estimate of groundwater carbon sources and sinks for global carbon budgets. Here we reveal the transformations of DOM in aging groundwater using ultra-high resolution mass spectrometry combined with radiocarbon dating. Long-term anoxia and a lack of photodegradation leads to the removal of oxidised DOM and a build-up of both reduced photodegradable formulae and aerobically biolabile formulae with a strong microbial signal. This contrasts with the degradation pathway of DOM in oxic marine, river, and lake systems. Our findings suggest that processes such as groundwater extraction and subterranean groundwater discharge to oceans could result in up to 13 Tg of highly photo-labile and aerobically biolabile groundwater dissolved organic carbon released to surface environments per year, where it can be rapidly degraded. These findings highlight the importance of considering groundwater DOM in global carbon budgets.

[1] Australian Nuclear Science and Technology Organisation (ANSTO), New Illawarra Rd, Lucas Heights, NSW 2234, Australia. [2] Connected Waters Initiative Research Centre, UNSW Sydney, Sydney, NSW 2052, Australia. [3] School of Civil and Environmental Engineering, UNSW Sydney, Sydney, NSW 2052, Australia. [4] Department of Earth, Ocean, and Atmospheric Science, Florida State University, Tallahassee, FL 32310, USA. [5] School of Biological, Earth and Environmental Sciences, UNSW Sydney, Sydney, NSW 2052, Australia. [6] National Marine Science Centre, Southern Cross University, Coffs Harbour, NSW 2450, Australia. [7] Mark Wainwright Analytical Centre, UNSW Sydney, Sydney, NSW 2052, Australia. [8] National High Magnetic Field Laboratory, Florida State University, Tallahassee, FL 32310-4005, USA. ✉email: lizam@ansto.gov.au

Groundwater plays an important role in the global water and carbon cycles[1–3]. It is estimated that almost 44 million km³ of groundwater could be stored in the upper 10 km of the continental crust[4]. This quantity exceeds the amount of water stored in ice sheets in Antarctica, Greenland, and glaciers combined (30.16 million km³)[5–7], and far exceeds the combined fresh water available in lakes, swamps, and rivers (0.19 million km³)[8]. Substantial groundwater contributions occur in approximately 40% of non-dam stream sites in the United States[9]. In coastal waters, the flux of groundwater to oceans via subterranean groundwater discharge (SGD) is estimated at $2.2–2.4 \times 10^{12}$ m³ year$^{-1}$ [10]. This influx of groundwater to surface waters can impact ecosystem processes and biota[11], increase bulk radiocarbon ($^{14}$C) dissolved organic carbon (DOC) age in streams[12,13] and lead to high apparent bulk $^{14}C_{DOC}$ ages of aquatic organisms[14]. The rate and extent of climate change, which is expected to have a significant impact on ecosystems, water and food availability, and human health, is influenced by the magnitude of greenhouse gas accumulation in the atmosphere[15]. Recent research suggests that groundwater is a significant source of inorganic carbon to surface environments, with studies showing that bicarbonate fluxes from groundwater are an important yet overlooked source of $CO_2$ to the atmosphere[16]. Currently, the flux of groundwater DOC to the oceans is unknown[17,18]. However, they could be significant given that groundwater DOC concentrations typically exceed those of coastal DOC concentrations[19]. Furthermore, the potential for groundwater DOC conversion to atmospheric $CO_2$ through degradation processes is poorly understood. This is partly due to the current lack of studies investigating the molecular transformation of DOM in groundwater systems, and a previous conceptual framework suggesting that highly aged DOM such as that found in deep groundwater should be recalcitrant (stable and unreactive) because more labile DOM is processed first.

DOM consists of tens of thousands of molecules primarily containing carbon (C), hydrogen (H), oxygen (O), nitrogen (N), and sulphur (S). Interactions with the environment such as mineralization by microbes, photodegradation through exposure to sunlight, additional input of local DOM sources, and removal of some formulae result in changes to the number and arrangement of atoms, thereby changing DOM reactivity in the environment. The rate and extent of DOM processing can be indirectly influenced by general water chemistry processes, such as high levels of water-rock interaction (thereby resulting in changes in pH and dissolved minerals[20,21]), or input of waters affected by agricultural and urban pollution[22,23]. For example, variations in water chemistry modulate bacterial community compositions which subsequently affects biodegradation pathways[24], the production and photolysis of $Fe^{3+}$-DOM complexes[25], or the oxidation of DOM by hydroxyl radicals which are produced at lower pH[26] and in the presence of nitrate and nitrite[27]. Also relevant to climate change is sea-level rise and over-extraction of groundwater in coastal areas, which can result in seawater intrusion into aquifers, and can have the effect of altering groundwater ionic strength[28]. Changes to ionic strength can affect the ability of DOM to adsorb to mineral surfaces[29–31]. Furthermore, seawater intrusion can influence terminal electron acceptor availability and microbial composition of groundwater[32,33] which may result in changes to the metabolic pathways used to mineralise DOM and altered mineralisation rates[34].

To date, most of our understanding on DOM cycling is based on non-groundwater oxic aquatic environments (e.g., lakes, rivers, and oceans)[35,36] where carboxylic-rich alicyclic molecules (CRAM) containing relatively intermediate H/C and O/C ratios tend to be stable over time[37–40]. DOM formulae are classified as CRAM if they contain double bond equivalent (DBE)/C ratios of 0.30–0.68, DBE/H ratios of 0.20–0.95 and DBE/O ratios of 0.77–1.75[39]. Flerus et al.[38] investigated the degradation of DOM in marine environments using Fourier-transform ion cyclotron resonance mass spectrometry (FT-ICR MS) and established a marine DOM degradation index ($I_{DEG}$) using the ratio of the intensity of ten molecular formula found to be highly significantly correlated with $\Delta^{14}C_{DOC}$ (‰) in the ocean. Subsequently, a larger group of CRAM molecules, termed the Island of Stability (IOS), was shown to increase in relative abundance in aging marine DOM[37]. Increasing relative abundance of molecules lying within the H/C and O/C bounds of the IOS have since been shown to correlate with increasing DOC age in other non-groundwater aquatic environments[36,41], thereby raising the question of whether the degradation trajectory of natural DOM towards intermediate H/C and O/C ratios may be consistent, irrespective of environmental conditions. This was further supported by observations of a cascade of degradation processes driving the accumulation of a stable background DOM of similar molecular formulae in marine and lake samples (i.e., molecular-level convergence)[35].

Redox conditions play an important role in the degradation of DOM[42], which in part is determined by groundwater turnover (i.e., inflow and outflow). In oxic environments, DOM degradation is kinetically controlled due to the high amounts of energy released from oxygen reduction[43]. Theoretically all DOM molecules are biodegradable in oxic environments[44] with preferential biodegradation of high H/C (>1.5) aliphatic, protein-like, lipid-like and heteroatom-containing formulae in surface waters such as rivers, lakes, and oceans[45–49]. Groundwater is often more reduced than other aquatic environments due to biotic factors such as the progressive microbial consumption of molecular oxygen and other terminal electron acceptors, and abiotic factors limiting oxidation, including the exclusion of sunlight which produces reactive oxygen species that can oxidize DOM[50]. Research has shown that hydroxyl radicals can also be produced non-photochemically in dark environments[51] due to the transfer of electrons from redox-active hydroquinone moieties to $O_2$[52], or due to Fe-mediated reduction of $O_2$[53,54]. In both cases molecular oxygen is required, thus dark and continuously reduced conditions would prevent these reaction pathways from occurring. Deoxygenation or removal of oxygenated DOM formulae from solution may also result from abiotic processes, such as the preferential adsorption of higher O/C ratio formulae to iron oxyhydroxides[55], or the hydrolyzation of diesters or beta-ketoesters by aqueous acid and the decarboxylation of the resulting diccid or beta-keto acid with heat[56]. In anaerobic environments, biotic degradation of DOM is thermodynamically controlled, favoring DOM preservation as a result of the high amount of energy required to consume organic carbon[44]. Under these conditions, preserved DOM often consists of formulae with low nominal oxidation state of carbon (NOSC < 0), including some amino acids, sugars and lipids, complex organics and membrane-type compounds[44]. Importantly, some low NOSC formulae can be aromatic and susceptible to photodegradation[57]. Hydrogenation of DOM can also occur in the absence of inorganic terminal electron acceptors, where unsaturated compounds may act as organic electron acceptors and serve as a $H_2$ sink[58]. In geological settings, such as deep groundwaters, $H_2$ may be made available through disassociation of water resulting from the radioactive decay of $^{238}$U, $^{232}$Th and $^{40}$K[59]. The hydrogenation of organic matter allows for high $CO_2:CH_4$ production ratios to persist in anaerobic environments[58], thereby likely resulting in increases in H/C and decreases in O/C ratios of DOM. In groundwater, residence times can be millions of years, photo-oxidative processes are absent, and anoxic conditions often prevail. These anoxic conditions result in the slow biodegradation of

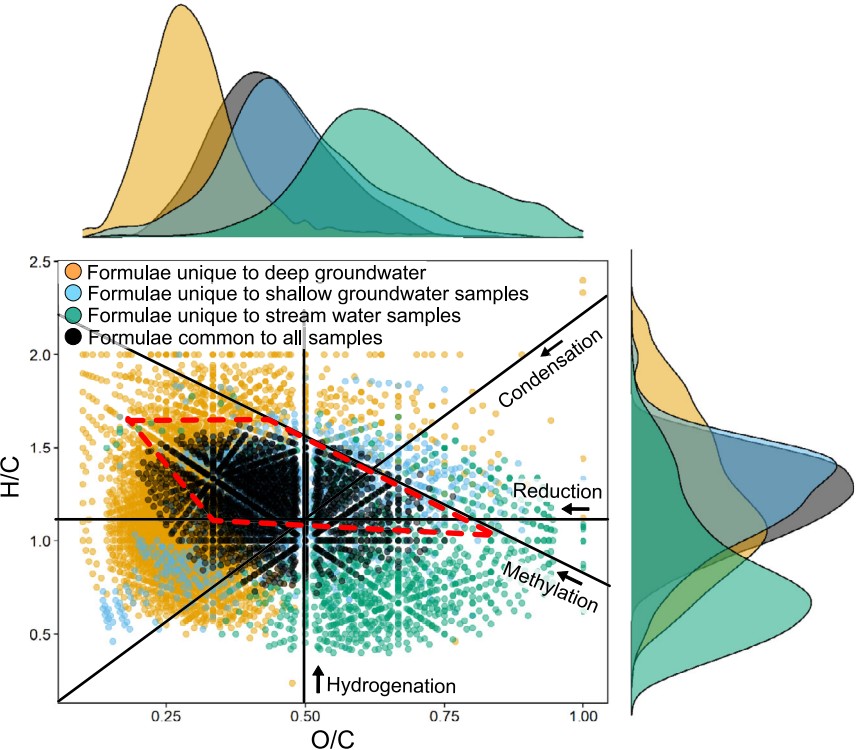

**Fig. 1 van Krevelen Diagram (VKD) with marginal density plots of molecular formulae.** VKD shows formulae unique to only deep groundwater (yellow), shallow groundwater (blue) or stream water (green), and molecular formulae common to all samples (black) as presented in Supplementary Table 4. Black solid lines represent the direction of change in H/C and O/C for chemical reactions including hydrogenation, methylation (or alkyl chain elongation), reduction, and condensation[112]. Marginal density plots are area-normalised and use a Gaussian kernel function for kernel density estimation. Individual VKDs for all samples are shown in Supplementary Figure 1. The approximate region where carboxylic-rich alicyclic molecules (CRAM) lie is represented by the red dashed polygon[57]. It is noted that CRAM formulae may represent a highly complex mixture of isomers[40]. As FT-ICR MS does not allow for discrimination between isomers, it is not possible to determine the chemical structure of these individual CRAM formulae.

DOM, which may additionally result in reduced degradation of microbial biomass and hydrogenated DOM formulae including biolabile high H/C aliphatics, peptides and sugars[60].

Here, we hypothesize that reduced and dark groundwater conditions allow for the build-up and potentially long-term storage of both photolabile low O/C and biolabile high H/C formulae, in contrast to sunlight-exposed and oxic aquatic environments. We examine stream water, shallow groundwater, and deep groundwater samples, representing a theoretical trajectory from freshly generated surface DOM to highly processed deep aquifer DOM. This allows conclusions to be drawn about DOM transformation as it enters and persists in confined deep aquifers. Radiocarbon ($^{14}$C) and high-resolution DOM characterisation techniques are used to resolve the groundwater DOM degradation trajectory and reveal for the first time, the molecular fingerprint of ancient DOM (up to $25,310 \pm 600$ years before present, BP) from a confined, methanogenic aquifer system.

## Results and Discussion

**Redox conditions, dissolved organic matter age and compositions.** The DOM molecular formulae of streams, shallow groundwater from unconfined aquifers (<41 m below ground surface (m bgs)), and deep groundwater from a confined aquifer (>500 m bgs) were highly divergent, as shown by the van Krevelen Diagram (VKD) in Fig. 1. Stream water samples from this study contained dissolved oxygen (DO) > 7 mg L$^{-1}$ (Supplementary Table 1), DOC concentrations between 1.34–8.26 mg L$^{-1}$ (Supplementary Table 2), and modern bulk $^{14}$C$_{DOC}$ (i.e., younger than the radiocarbon reference year of 1950 CE). Local and regional shallow groundwater samples ($n = 9$) either contained detectable levels of dissolved

oxygen (DO) and nitrate (NO$_3^-$) or manganese (Mn$^{2+}$), ferrous iron (Fe$^{2+}$) or sulfate (SO$_4^{2-}$)[61] (Supplementary Table 1). These shallow groundwater samples contained low DOC concentrations (0.48–1.58 mg L$^{-1}$, Supplementary Table 2), intermediate to recent dissolved inorganic carbon (DIC) ages of 4000 years BP to modern, and DOC ages of 2540 years BP to modern (Supplementary Table 3). The deep confined groundwater samples were low in DO, NO$_3^-$, Mn$^{2+}$, SO$_4^{2-}$ and Fe$^{2+}$ (Supplementary Table 1), and experienced methanogenic conditions[62] with low DOC concentrations (0.75–1.10 mg L$^{-1}$). These deep groundwater samples had DIC ages of >50,000 years BP and DOC ages of 19,080–25,310 years BP (Supplementary Table 3).

**Dissolved organic matter processing in groundwater.** In shallow groundwater, DOM undergoes transformation through the production of low O/C CRAM formulae and removal of low H/C aromatic formulae (Fig. 2, Supplementary Fig. 2) whilst deep methanogenic groundwaters show an increase in aliphatics and aromatic low O/C (<0.5) formulae (Fig. 2). An initial removal of aromatic groups from the stream water samples during processing in shallow groundwater, and the accumulation of these formulae in deep methanogenic groundwater is evidenced through double bond equivalent (DBE)/O slopes of 0.71 ($R^2 = 0.98$, $p = 2.7 \times 10^{-14}$) and 1.07 ($R^2 = 0.98$, $p = 2.7 \times 10^{-14}$) respectively (Fig. 2a, d). As carboxyl groups consist of one DBE and two O atoms, a linear regression slope for DBE/O of approximately 0.5 is indicative of DOM containing carboxyl groups[63]. Slopes greater than 0.5, as observed particularly in the stream water and deep groundwater (Fig. 2a, d), suggest the presence of more cyclic bonds[63]. Adsorption to mineral surfaces

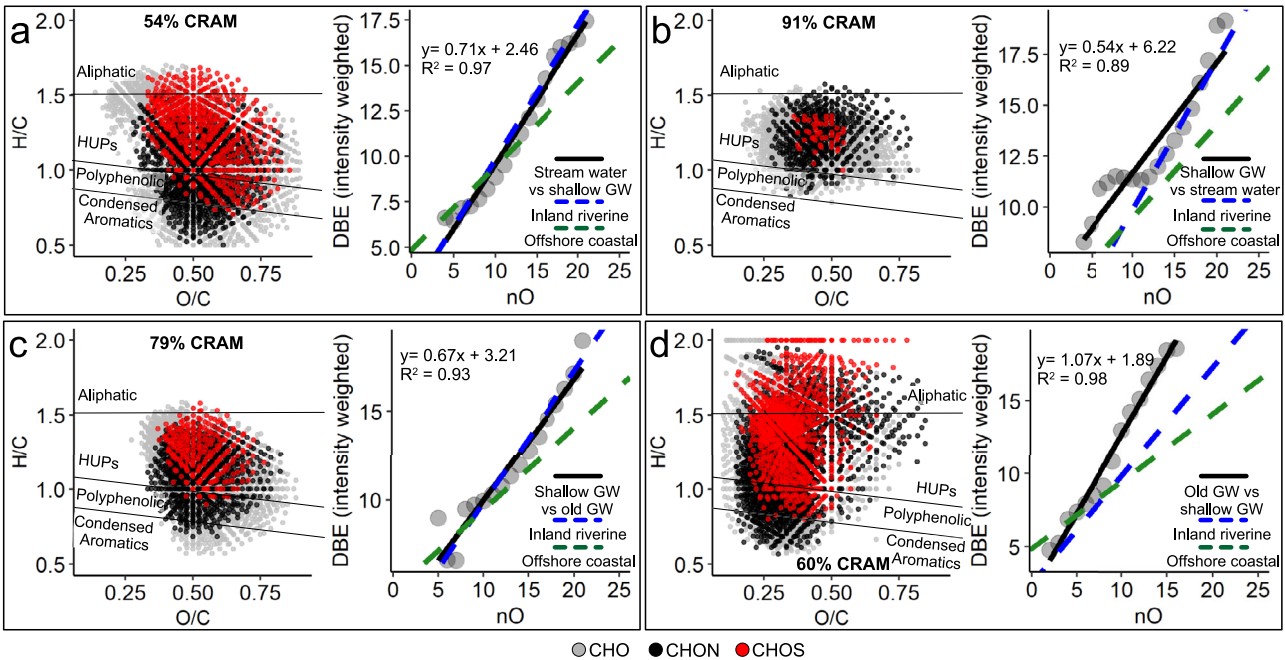

**Fig. 2 Comparison of dissolved organic matter (DOM) formulae in stream water, shallow groundwater and deep confined groundwater.** van Krevelen Diagrams (VKD's; H/C vs O/C ratios) are shown on the left of each panel, whilst regression lines (solid black lines) showing the correlation between intensity weighted average double bond equivalent (DBE) and number of oxygen atoms in each molecule for formulae: **a** higher in median relative intensity (intensity difference <0) in stream water compared to shallow groundwater, **b** higher in median relative intensity (intensity difference <0) in shallow groundwater compared to stream water, **c** higher in median relative intensity (intensity difference <0) in shallow groundwater compared to deep groundwater, and **d** higher in median relative intensity (intensity difference <0) in deep groundwater compared to shallow groundwater (see also Supplementary Table 5). Green and blue dashed lines represent the DBE/O regression lines for offshore coastal DOM and inland riverine DOM[63], respectively. Point colours in the VKD's correspond to CHO (grey), CHON (black) and CHOS (red) formulae. "HUPs" refers to highly unsaturated and phenolic formulae. Note: the median differences in molecule intensities have been used in the VKDs to reduce the influence of outliers with very high or low compound intensity. Percentages of carboxylic-rich alicyclic molecules (CRAM) are weighted based on the total sum intensity of the CRAM molecules divided by the sum intensity of all molecules shown in the VKD. A comparison of intensity differences based on molecular mass is provided in Supplementary Figure 3. Separate VKDs are shown for each compound class (CHO, CHON and CHOS) in Supplementary Figure 4.

is a likely removal mechanism for these formulae in shallow groundwater[55]. A subsequent production or retention of low O/C CRAM formulae in shallow groundwater is revealed by the overall high percent relative abundance (% RA) of CRAM (91% RA, Fig. 2b), DBE/O slope of 0.54 (Fig. 2b), and the increase in CRAM % RA with $\Delta^{14}C_{DOC}$ (‰) observed in the shallow groundwater samples (Supplementary Figure 2a). These CRAM formulae contain a lower average weighted O/C ratio (0.47) compared to CRAM that are higher in intensity in stream water samples (0.53).

Marine and soil bacterial metabolites and by-products display high H/C and low O/C[60,64,65], and can result in an input of N and S containing DOM[66,67] which are rapidly aerobically biodegraded[68]. Deep groundwater samples from this study are similarly characterised by aerobically biolabile high H/C (1.5 > H/C < 2.0) aliphatic and peptide-like formulae, low O/C (<0.50) formulae, and heteroatom (N or S) containing DOM. These formulae account for 30.3%, 79.9% and 22.0% of the formulae that are higher in intensity in deep groundwater compared to shallow groundwater, respectively (Fig. 2d). Some of the low O/C formulae in Fig. 2d also contain low H/C and are classified as condensed aromatic or polyphenolics (Supplementary Table 5). Aromatic and polyphenolic formulae are sometimes associated with terrestrial inputs from vascular plants in rivers and streams[69,70], sedimentary inputs in the subsurface[71] (see Supplementary Note 1), or condensation reactions[72]. Fluorescence-derived parameters such as Biological Index (BIX), Fluorescence Index (FI) and Peak T, which can be used as indicators of recent microbial activity[73,74], suggest a

predominantly microbial source of these formulae in the deep groundwater samples. A negative correlation is observed between BIX and $\Delta^{14}C_{DOC}$ (‰) ($p = 2.22 \times 10^{-2}$, Supplementary Figure 5), indicating greater bacterial DOM contribution as groundwater DOM ages. This is supported by a high FI and tryptophan-like DOM (Peak T) in deep groundwater samples (Supplementary Figure 6 for flouresence excitation emission matrices and Supplementary Table 2). As previously noted, high H/C aliphatics, peptide-like and heteroatom containing formulae have a high biolability in aerobic environments. Their high abundance in deep confined groundwaters therefore suggests a slow microbial recycling of DOM into increasingly high H/C and N or S containing formulae rather than the preservation of aliphatic or heteroatom containing molecules from an original terrestrial source. These formulae are preserved in the methanogenic groundwaters along with photodegradable low O/C aromatic formulae[57] which are protected from sunlight in aquifers. Furthermore, LaRowe and Van Cappellen[44] demonstrated that the degradation of organic molecules with low nominal oxidation state of carbon (NOSC, approximately <0) is thermodynamically inhibited under reduced conditions, thus, biodegradation of the low O/C formulae in the deep groundwater samples would be limited in anoxic waters due to their low weighted average NOSC (−0.12).

**Degradation in groundwater vs. surface aquatic environments.** Significantly higher median H/C and lower median O/C ratios (both $p = 2.2 \times 10^{-16}$, Fig. 3a, b) are observed in groundwater

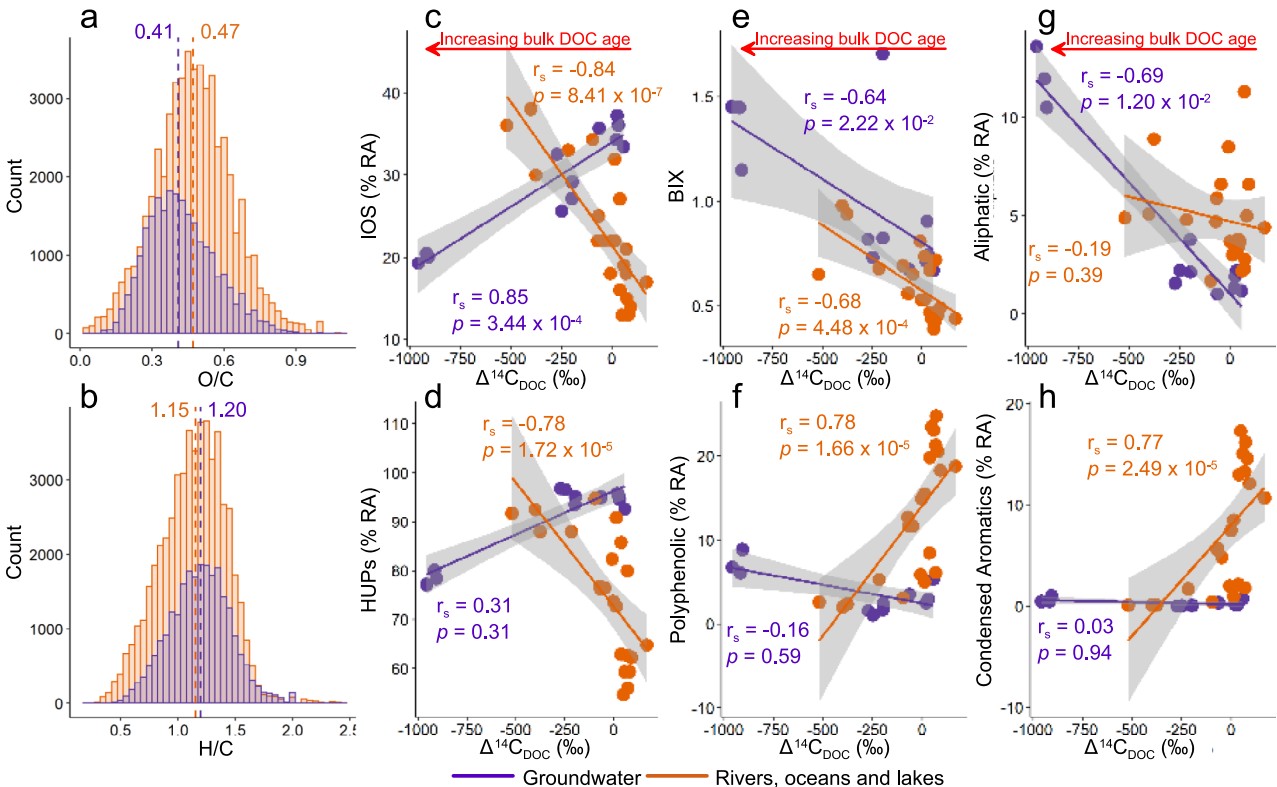

**Fig. 3 Comparison between groundwater and non-groundwater (river, ocean and lake) dissolved organic matter (DOM) composition. a, b** show histograms of the O/C and H/C of formulae identified in a range of non-groundwater environments (orange) including Swedish[105] and German[35] lakes, Mediterranean Sea[35,74], North Sea[35,113], Antarctic bottom water[35], North Atlantic deep water[35], Suwannee River[114], and streams from this study (Macquarie River, Bell River and Elfin Crossing samples). Groundwater DOM formulae identified in shallow and deep groundwater from this study are shown in purple. Comparisons of the numbers of H, C and O in each environment are shown in Supplementary Fig. 7. A comparison between percent relative abundance (% RA) of Island of Stability (IOS) formulae, % RA highly unsaturated and phenolic compounds (HUPs), biological index (BIX), % RA polyphenolic, % RA aliphatic and % RA condensed aromatics vs $\Delta^{14}C_{DOC}$ (‰) in non-groundwater samples from Kellerman, et al.[36] are shown in **c–h** with Spearman rho and p-values represented by $r_s$ and p respectively. Dashed lines in **a, b** indicate median values in non-groundwater (orange) and groundwater (purple) environments. F test to compare variances reveals that the variances are significantly different ($p < 2.2 \times 10^{-16}$), with Wilcoxon tests showing significantly lower and higher median O/C and H/C in groundwater compared to non-groundwater samples respectively (both $p = 2.2 \times 10^{-16}$).

compared to a range of non-groundwater aquatic environments including lakes, oceans, rivers, and streams. A progressive shift towards low O/C and higher H/C from stream water to shallow groundwater and deep groundwater is confirmed by significant positive and negative relationships between $\Delta^{14}C_{DOC}$ (‰) and weighted average O/C ($p = 2.2 \times 10^{-16}$) and H/C ($p = 2.4 \times 10^{-4}$), respectively (Supplementary Figure 5). This corresponds to a significant decrease in the % RA of IOS formulae in groundwater over time ($p = 3.4 \times 10^{-4}$). These findings contrast marine waters where decreases in H/C and increases in NOSC have been reported with increasing DOM residence times[37,38], and are associated with an overall increase in IOS formulae over time in non-groundwater environments[36,37,41] ($p = 5.0 \times 10^{-7}$, Fig. 3c).

Increases in IOS formulae in non-groundwater aquatic environments likely result from kinetic controls on DOM degradation under oxic conditions. In methanogenic portions of the ocean, such as the oxygen minimum zone or in anoxic marine sediments, there may be a build up of low O/C and high H/C formula similar to what is observed in deep groundwaters, due to thermodynamic constraints on DOM degradation. For example, Gan, et al.[75] demonstrated an increase in low O/C and high H/C formulae in marine sediments from the Western Mediterranean Sea when exposed to methanogenic conditions for approximately 38 days. Slower respiration rates thus appear to drive the

preferential biodegradation of high NOSC formulae and preservation of low NOSC DOM and aerobically biolabile high H/C microbial metabolites and biomass in groundwater[76] (Fig. 4). In contrast, DOM in rivers, oceans and lakes typically represents a continual mixing of fresh DOM sources from primary production with pre-existing DOM produced from biodegradation and photodegradation in the primarily oxic conditions. This mixing governs the average DOM composition and results in an overall accumulation of higher NOSC and IOS formulae. Our findings therefore indicate that the stability of the IOS may be context-dependent, relevant in surface environments exposed to photo-irradiation and/or oxic or fluctuating redox conditions, but not relevant to dark and reduced groundwater environments.

**Implications**. Our results show that the current paradigm of aged, highly processed, apparently stable DOM occurring in the centre of H/C versus O/C space may actually be constrained to well-mixed, oxic, open waters, and that the oldest DOM appears instead to occur in dark anoxic aquifers where molecular oxygen and attendant reactive oxygen species are unavailable. In these environments, the most persistent formulae are biolabile with high H/C, or contain low O/C including aromatic formulae. Notably, many low O/C aromatic formulae are readily converted into oxidised or aerobically biolabile aliphatic formulae with

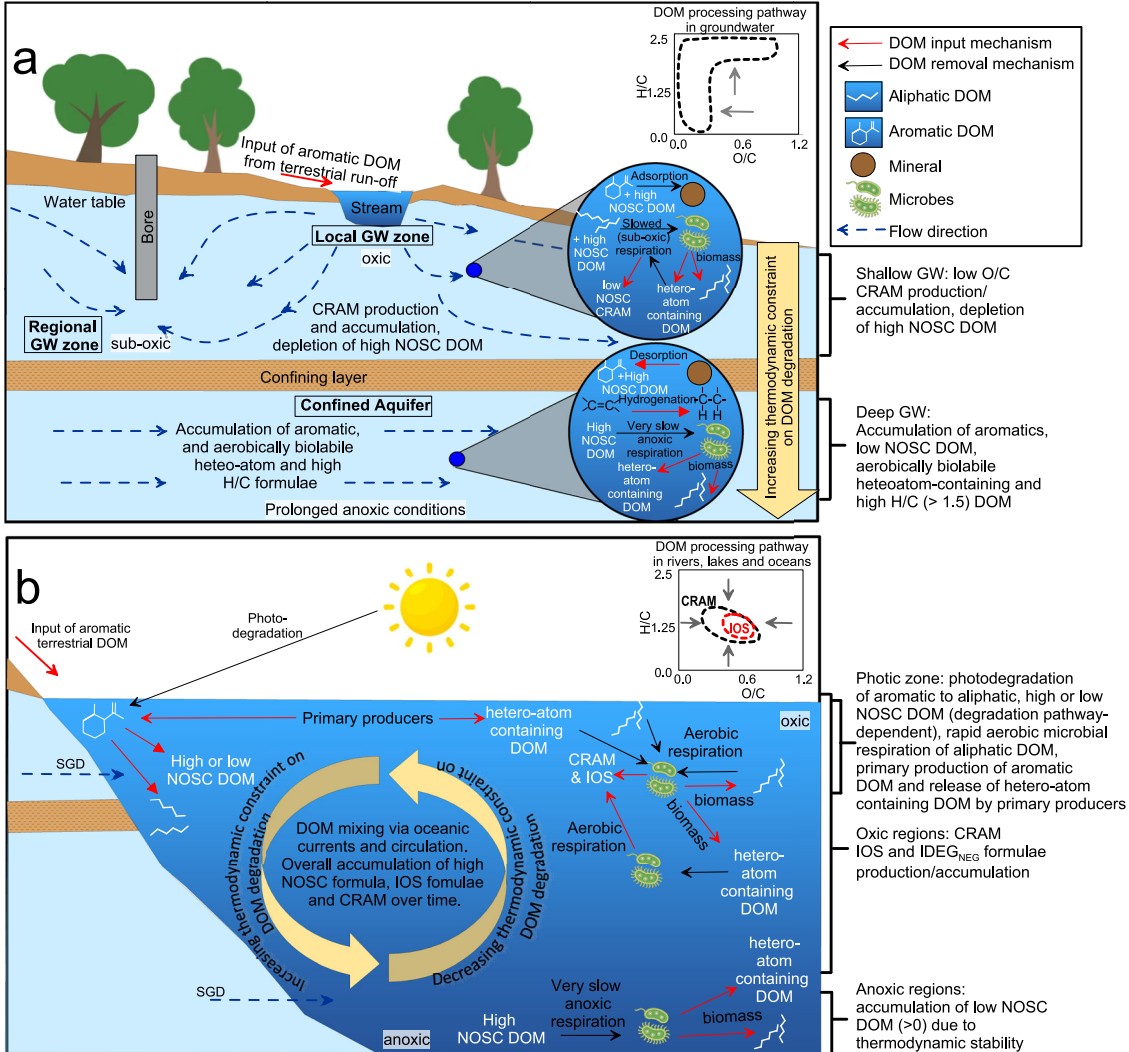

**Fig. 4 Contrasting dissolved organic matter (DOM) processing pathways in groundwater and non-groundwater (river, lake and ocean) environments. a** DOM in shallow groundwater is removed following exposure of hetero-atom containing DOM and DOM with high nominal oxidation state of carbon (NOSC) to biodegradation in suboxic conditions, and the adsorption of aromatic and high NOSC formulae. Accumulation of low O/C, non-Island of Stability (IOS) carboxylic-rich alicyclic molecules (CRAM) occurs in this region. In deep groundwater where conditions are strongly anoxic, respiration rates are low and aerobically biolabile aliphatic and hetero-atom containing DOM are accumulated from microbial biomass in conjunction with a preferential removal of high O/C DOM. Hetero-atom containing DOM may be further accumulated through the hydrogenation of unsaturated formulae and inorganic sulfurization of lipids[115]. Aromatic DOM also accumulates from microbial biomass with possible contribution from DOM desorbed under high pH[116]. These processes result in an overall decrease in O/C and an increase in H/C with DOC age. Dark blue dashed arrows denote flow direction. Groundwater DOM processing contrasts with processing of DOM in rivers, lakes and oceans (**b**) where primary production and subsequent photodegradation of aromatic formulae results in either low or high O/C formulae, and high H/C formulae such as aliphatics which are then rapidly biodegraded. Primary producers may also release biolabile hetero-atom containing DOM[68]. Any build up of DOM in an anoxic pocket of seawater which is biolabile to aerobic microbes would be circulated and biodegraded upon return to oxic conditions. Exposure to sunlight and the primarily oxic conditions of marine environments result in an overall decrease in DOM H/C, and increase in O/C and IOS formulae with DOC age. Processing pathways for both environments are shown in van Krevelen Diagrams in upper right corners of each panel.

exposure to photoirradiation[57]. Natural processes and anthropogenic activities can transport large quantites of groundwater to surface environments where it is exposed to sunlight and oxygen. For example, subterranean groundwater discharge results in the movement of 2397 km³ year⁻¹ of groundwater into coastal marine environments[10]. Additionally, global groundwater extraction for industrial, domestic and agricultural usage is estimated at 982 km³ groundwater year⁻¹ [77]. A large proportion of this is is expected to be ancient DOM with up to 85% of total groundwater in the upper 1 km of continental crust shown to be recharged by precipitation >12,000 years ago[78]. Using the global median and mean groundwater DOC concentrations of 1.2 mg C L⁻¹ and 3.8 mg C L⁻¹ [79] respectively, these values represent approximately 2.9–9.1 Tg of groundwater DOC exported to oceans from subterranean groundwater discharge annually, and approximately 1.2–3.7 Tg of groundwater DOC removed during groundwater extraction annually. This equates to total DOC fluxes of between 4.1–12.8 Tg year⁻¹ associated with these processes, with the lower estimate approximately equivalent to twice the amount of DOC exported from the Mississippi River each year (2.10 Tg), and the upper estimate approximately equivalent to the amount of DOC exported annually from the Congo River (12.40 Tg)[80]. Rates of groundwater extraction are expected to rise throughout the 21st century, with annual global extraction

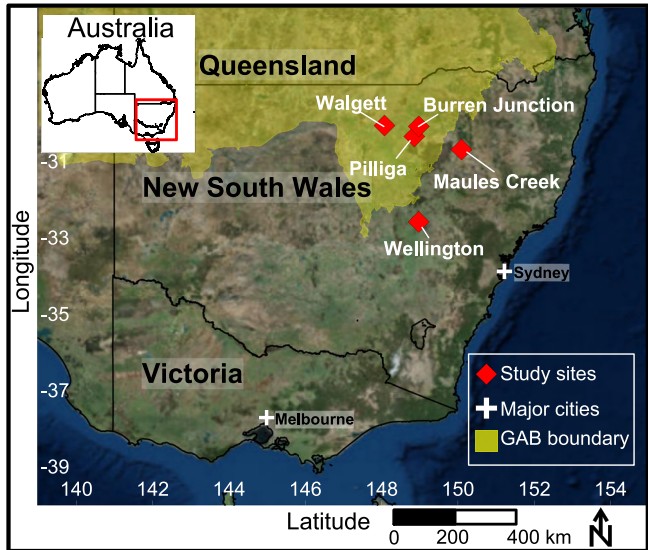

**Fig. 5 Sampling sites. Sampling locations are shown as red diamonds, with the Great Artesian Basin boundary shown in yellow.** Major cities are shown as white crosses.

projected at 1621 km³ by 2099[81], 61% higher than the value used here for estimating DOC from groundwater extraction. If exported from ancient groundwaters, this DOM would be highly biolabile and photolabile upon transport to surface environments. Our results therefore confirm that groundwater may be an important source of labile DOM upon transfer to surface environments, and highlights the importance of it's inclusion in global carbon budget estimates.

## Methods

**Sampling**. Sampling for shallow groundwater was performed at Maules Creek and Wellington, Australia (see Fig. 4 and for further context Supplementary Information in McDonough, et al.[82]). These shallow groundwater samples were collected from the hyporheic zone or regional unconfined aquifers, and are all known to be recharged by nearby streams during floods or dam releases[82–84]. Stream samples included Bell River and Macquarie River at Wellington, and Elfin Crossing on Maules Creek. Further information regarding the connection between surface and groundwater at Wellington and Maules Creek and the climatic conditions at the time of sampling for Bell River, Macquarie River, Elfin Crossing, WRS03, WRS05, BH12-4, BH18-2, EC31, EC3 and EC6 is also provided in McDonough, et al. and the associated Supplementary Information. BH17-2 and BH17-4 were sampled at Middle Creek, a tributary to Maules Creek. Deep groundwater was sampled from Walgett, Pilliga and Burren Junction within a confined thermogenic aquifer which forms part of the Great Artesian Basin, underlying 22% of the continent of Australia (Fig. 5). In total, three stream water samples, nine shallow groundwater samples (<41 m below ground surface (m bgs)) and three deep groundwater sample (>500 m bgs) were collected.

Further information regarding sample locations and depths are provided in Supplementary Table 1. Low DOC concentrations are noted in two of the stream water samples (Bell River (1.87 mg C L⁻¹) and Elfin Crossing (1.34 mg C L⁻¹)) compared to the Macquarie River (8.26 mg L⁻¹). Elfin Crossing stream water originates from an upstream spring[85]. This water has therefore been processed in the subsurface for a period of time before briefly re-emerging as surface water. Keshavarzi, et al.[84] similarly show that Bell River water has been transported through the karstic limestone of the Wellington region.

All sites were sampled for bulk DOC concentration, liquid chromatography—organic carbon detection (LC-OCD), Fourier transform ion cyclotron resonance mass spectrometry (FT-ICR MS), cations, anions, alkalinity, stable water isotopes, radiocarbon in DOC ($^{14}C_{DOC}$) and dissolved inorganic carbon ($^{14}C_{DIC}$), stable carbon isotopes in DOC ($^{13}C_{DOC}$) and DIC ($^{13}C_{DIC}$) using the sampling equipment, bottles and methods outlined in McDonough et al[82].

Field parameters including dissolved oxygen (DO), pH, and electrical conductivity (EC) were measured before and during sampling using two HACH HQ40D multimeters attached to an in-line Sheffield flow-cell. Deep groundwater samples at Pilliga, Walgett, and Burren Junction were collected from artesian free-flowing bores after field parameters stabilised. Samples were collected directly from a 0.45 μm in-line fast flow filter (Waterra) connected to tubing which was attached to the bore sample taps. Stream water and shallow groundwater samples (<40 m

bgs) were collected using the methods and equipment outlined in McDonough et al[82]. Alkalinity was measured through gran-titrations using sulfuric acid ($H_2SO_4$) within 12 hours of sample collection. Cation samples were acidified with 65% Suprapur® nitric acid within 12 hours of sample collection.

**Pre-treatment, analysis, and post-processing**. Pre-treatment methods for $^{14}C_{DOC}$ and $^{14}C_{DIC}$ are outlined in McDonough et al[82]. $CO_2$ extraction and graphitisation procedures are outlined in Hua et al.[86] $^{14}C_{DOC}$ (pMC) results were converted to $\Delta^{14}C_{DOC}$ (‰) using the methods described in International Atomic Energy Agency[87]. $\Delta^{14}C_{DOC}$ (‰) values represent a per mille depletion or enrichment in $^{14}C$ relative to the standard, normalised for $^{13}C$ isotopic fractionation. This unit has been used to allow for comparison to other non-groundwater aquatic DOM degradation studies which predominantly report DOC age as $\Delta^{14}C_{DOC}$ (‰). Solid phase extraction (SPE) was performed prior to analysis by FT-ICR MS using reversed phase BondElut PPL sorbent (100 mg cartridge, Agilent Technologies) using the methods in Dittmar et al[88]. A detailed comparison of potential differences resulting from comparing bulk DOM LC-OCD and bulk DOM $^{14}C_{DOC}$ results with SPE DOM FT-ICR MS results are provided in the Supplementary Information of McDonough et al[82].

A summary of the analytical instruments and laboratories where analyses were performed for each analysis type are provided in Table 1.

For FT-ICR MS analysis, the sample solution was infused via a microelectrospray source[89] (50 μm i.d. fused silica emitter) at 500 nL/min by a syringe pump. Typical conditions for negative ion formation were: emitter voltage, −2.4–2.9 kV; S-lens RF level: 45%; and heated metal capillary temperature, 350 °C. DOM extracts were analyzed with a custom-built hybrid linear ion trap FT-ICR mass spectrometer equipped with a 21 T superconducting solenoid magnet[90,91]. Ions were initially accumulated in an external multipole ion guide (1–5 ms) and released $m/z$-dependently by decrease of an auxiliary radio frequency potential between the multipole rods and the end-cap electrode[92]. Ions were excited to $m/z$-dependent radius to maximize the dynamic range and number of observed mass spectral peaks (32–64%)[93], and excitation and detection were performed on the same pair of electrodes[94]. The dynamically harmonized ICR cell in the 21T FT-ICR is operated with 6 V trapping potential[95,96]. Time-domain transients of 3.1 s were acquired with the Predator data station that handled excitation and detection only, initiated by a TTL trigger from the commercial Thermo data station, with 100 time-domain acquisitions averaged for all experiments[97]. Mass spectra were phase-corrected and internally calibrated with 3–5 highly abundant homologous series that span the entire molecular weight distribution based on the "walking" calibration method[98]. Experimentally measured masses were converted from the International Union of Pure and Applied Chemistry (IUPAC) mass scale to the Kendrick mass scale[99] for rapid identification of homologous series for each heteroatom class (i.e., species with the same $C_cH_hN_nO_oS_s$ content, differing only by degree of alkylation)[100]. For each elemental composition, $C_cH_hN_nO_oS_s$, the heteroatom class, type (double bond equivalents, DBE = number of rings plus double bonds to carbon, DBE = $C - h/2 + n/2 + 1$)[101], and carbon number, c, were tabulated for subsequent generation of heteroatom class relative abundance distributions and graphical relative-abundance weighted images and VKDs. Peaks with signal magnitude greater than 6 times the baseline root-mean-square (rms) noise at $m/z$ 500 were exported to peak lists, and molecular formula assignments and data visualization were performed with PetroOrg © software[102]. Molecular formula assignments with an error >0.5 parts-per-million were discarded.

Molecules were classified as CRAM where DBE/C were between 0.30-0.68, DBE/H were 0.20-0.95 and DBE/O were between 0.77-1.75 as per Hertkorn, et al.[39]. A modified aromaticity index ($AI_{mod}$) was calculated for individual molecular formulae per the methods of Koch and Dittmar[103] and Koch, et al.[104]. Formulae with $AI_{mod} \leq 0.5$, $0.5 < \leq 0.66$, and >0.66 are defined as highly unsaturated and phenolics, polyphenolic and condensed aromatic respectively. Formulae with $1.5 \leq H/C \leq 2.0$, $O/C \leq 0.9$ and $N = 0$ are defined as aliphatic. Formulae with O/C > 0.9 are defined as sugar-like whilst peptide-like are defined as $1.5 \leq H/C \leq 2.0$, and $N > 0$[105]. IOS compounds listed in the Appendix of Lechtenfeld, et al.[37] were matched to compounds within the current dataset and weighted by summing the relative intensity of IOS molecules within each sample and dividing by the sum of the relative intensities of all molecules within the sample. Degradation index ($I_{DEG}$) values were calculated as per Flerus, et al.[38]. Figure 2 was prepared using ggplot2[106] after calculating the difference in median intensity value of each molecule in stream water (Macquarie River, Bell River, and Elfin Crossing), shallow groundwater (BH17-2, BH17-4, BH12-4, BH18-2, EC31, EC6, EC3, WRS03, and WRS05) and deep confined groundwater (Pilliga, Burren Junction and Walgett). Average weighted DBE vs number of oxygen (nO) atoms were calculated by multiplying the DBE value for each molecular formula by the difference in intensity observed between stream water, shallow groundwater, and deep groundwater for the same molecular formula. The sum of the values within each category of nO (i.e., 2–21) was then divided by the sum of intensity differences within each category of nO to obtain the weighted average DBE for each nO. Spearman correlations presented in Fig. 3 were prepared in R v.1.1.456 using the ggplot2 library.

Post-processing for LC-OCD involved the use of ChromCALC (DOC-LABOR, Karlsruhe, Germany) which allows for chromatogram visualisation and peak fitting for DOC fractions. Hydrophillic fractions are assigned based on their molecular

**Table 1 Analytical instruments and analysis locations.**

| Analysis type | Analytical instrument | Analysis location |
| --- | --- | --- |
| Cations | Perkin Elmer NexION 300D inductively coupled plasma (ICP) mass spectrometer and Perkin Elmer Optima 7300 ICP optical emission spectrometer | Mark Wainwright Analytical Centre (UNSW Sydney) |
| Anions | Dionex Ion Chromatography System with an IonPac AS14A analytical column | Mark Wainwright Analytical Centre (UNSW Sydney) |
| N-NO$_3$ and N-NH$_4$ | Lachat flow injection analyser | Mark Wainwright Analytical Centre (UNSW Sydney) |
| $\delta^2$H and $\delta^{18}$O | Off-Axis Integrated Cavity Output Spectroscopy using an LGR Liquid Water Isotope Analyser[111] | UNSW Sydney |
| $\delta^{13}C_{DIC}$ | Delta V Advantage Isotope Ratio Mass Spectrometer | Australian Nuclear Science and Technology Organisation (ANSTO) |
| $\delta^{13}C_{DOC}$ | O.I. Analytical Model 1030 Total Organic Carbon Analyzer interfaced to a PDZ Europa 20-20 isotope ratio mass spectrometer utilizing a GD-100 Gas Trap Interface. | UC Davis Stable Isotope Facility, California |
| $^{14}C_{DOC}$ | Accelerator Mass Spectrometry | Australian Nuclear Science and Technology Organisation (ANSTO) |
| $^{14}C_{DIC}$ | Accelerator Mass Spectrometry | Australian Nuclear Science and Technology Organisation (ANSTO) |
| Total DOC | Aurora 1030 wet oxidation TOC analyser | Mark Wainwright Analytical Centre (UNSW Sydney) |
| Fluorescence | Horiba Scientific Aqualog fluorometer | UNSW Sydney |
| LC-OCD | DOC-LABOR LC-OCD size-exclusion chromatography | Mark Wainwright Analytical Centre (UNSW Sydney) |
| FT-ICR MS | Negative mode (−) electrospray ionisation (ESI) on a 21 Tesla super conducting magnet (Bruker, U.S.)[90,91] | National High Magnetic Field Laboratory, Tallahassee, Florida |

weights which are determined by retention times. Hydrophilic DOC is calculated as the concentration of DOC that passes through the column, whilst hydrophobic DOC (HOC) is defined as the portion of DOM that is retained on the column. HOC (% RA) is calculated by the DOC concentration of the sample minus the concentration of hydrophilic DOC. Further details can be found in Huber, et al[107].

Fluorescence data was corrected for inner filter effects, 1st, and 2nd order Rayleigh scattering, and normalised to Raman Units using the area of the Raman peak (measured each day) in the Aqualog software after sample measurement. Post-processing to calculate peak T, biological index (BIX), and fluorescence index (FI) was performed in R v.1.1.456 using the packages staRdom and eemR. T represents tryptophan-like (ex: 275, em: 340) peaks respectively[108]. BIX provides information regarding recent autochthonous DOM contribution and is calculated as the fluorescence at an excitation of 310 nm and emission at 380 nm, divided by the fluorescence at excitation 310 nm and emission at 430 nm[109]. FI is calculated as the ratio of fluorescence at emission 450 nm and 500 nm obtained at an excitation of 370 nm. Higher FI values (~1.9) indicate microbially derived fulvic acids (FA), whilst lower values (~1.4–1.5) indicate terrestrially derived FA[110]. Higher BIX values (>1) represent higher amounts of microbially-derived proteins.

**DOC release estimations and uncertainties**. DOC release estimations from SGD and global groundwater extraction were obtained from Zektser, et al.[10] and Margat, and van der Gun[77] respectively. Zektser, et al.[10] use a hydrologic-hydrogeodynamic method to estimate SGD. The calculation uses the average total thickness (200 m) of all aquifer systems from which groundwater is discharged to seas, the corresponding active porosity of rocks (0.1) in this zone, the average hydraulic gradient (0.005), and a 10 m day$^{-1}$ coefficient of filtration. They estimate a shoreline length of 600,000 km. Four rock types are defined for the purposes of the calculations: sedimentogenic-porous, sedimentogenic-fissured, karstic and magmatogenic-metamorphic. Notably, two different methods were used to compare the values of the total groundwater discharge to oceans – hydrogeodynamic (2200 km$^3$ year$^{-1}$) and hydrologic-hydrogeological (2400 km$^3$ year$^{-1}$). Both show similar results, however hydrogeodynamic methods require initial filtration parameters which are often not available due to poor knowledge of hydrogeological conditions in many areas. For this reason, the 2400 km$^3$ year$^{-1}$ value is taken to be more accurate. Full details of assumptions and uncertainties are provided in Zektser, et al.[10]. Calculations of DOC extracted each year globally from groundwater were made based on the sum of the continental groundwater extraction values for irrigation, domestic and industrial uses provided in Margat and van der Gun[77]. The global annual groundwater extraction value (982 km$^3$ year$^{-1}$) was summed with the global annual SGD value (2400 km$^3$ year$^{-1}$) and multiplied by the estimated global mean and median groundwater DOC concentrations presented in McDonough, et al.[79] (3.8 mg C L$^{-1}$ and 1.2 mg C L$^{-1}$). Global groundwater DOC concentrations are based on 9404 samples from 32 countries in 6 continents. The data show differences in DOC concentrations with changes in climate, land use, inorganic water chemistry (redox conditions), and groundwater residence times. The character of groundwater DOM is also likely affected by these parameters; thus, groundwater

DOM lability will differ depending on factors which include, but are not limited to, terminal electron acceptor concentrations, water residence time (and duration of exposure to anoxic conditions), infiltration rates, water temperature, and landuse. McDonough, et al.[79] identified significantly lower DOC concentrations in deep aquifers compared to shallow aquifers in a groundwater DOC dataset from the United States, suggesting that the median and mean global groundwater DOC concentrations may not be applicable when assessing the likely DOC flux from deep aquifers. We also note that whilst SGD occurs along the coast, samples in the study are taken from both inland and coastal regions, therefore the average climate types and land uses inherent in the 9404 groundwater sample locations may not represent the dominant land use and climate types in coastal areas. Finally, not all SGD will be from ancient or anoxic groundwaters which show the most photo-degradable and biolabile character.

## Data availability

The FT-ICR MS data generated in this study have been deposited in the Open Science Framework database under https://doi.org/10.17605/OSF.IO/94WFQ (https://osf.io/94wfq/). The FT-ICR MS data are publicly available. The inorganic chemistry and isotopic data for each sample are provided in tables within the Supplementary Information file.

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

## Acknowledgements

The authors thank Alan Williams, Fiona Bertuch, Simon Varley and Shwaron Kumar for their assistance with sample pre-processing for $^{14}C_{DIC}$ and $^{14}C_{DOC}$. The authors also thank Chris Dimovski for his assistance with $^{13}C_{DOC}$ samples, Kellie-Anne Farrawell and Barbora Gallagher for their assistance with tritium and $^{13}C_{DIC}$ analyses, and Narelle Hegarty for reviewing the manuscript. The authors would also like to thank Khorshed Chinu and staff from the Mark Wainwright Analytical Centre at UNSW Sydney for the analysis of inorganic chemistry and LC-OCD samples. The authors acknowledge Colin Gatgens, Bill Schwieger, Mick Dowell, and Evan Harris for their assistance with bore access at Burren Junction, Walgett and Pilliga. The authors thank Chad R. Weisbrod, John P. Quinn, Greg T. Blakney and Christopher L. Hendrickson for support and maintenance of the 21 T FT-ICR MS instrument. A portion of this work was performed at the National High Magnetic Field Laboratory, which is supported by the National Science Foundation Division of Materials Research and Division of Chemistry through DMR-1644779, and the State of Florida.

This research was funded by the Australian Research Council under Discovery Project DP160101379 (awarded to A.B., M.A., D.O. and K.M.). The authors acknowledge the financial support of the Centre for Accelerator Science at ANSTO through the Australian National Collaborative Research Infrastructure Strategy (NCRIS). Groundwater sampling was possible at Wellington and Maules Creek through the NCRIS Groundwater Infrastructure Project. The National High Magnetic Field Laboratory ICR User Facility is supported by the National Science Foundation Division of Chemistry through DMR-1644779, DMR-1157490 and the State of Florida. The National High Magnetic Field Laboratory ICR User Facility is supported by the National Science Foundation Division of Chemistry and Division of Materials Research through DMR-1644779, DMR-1157490 and the State of Florida.

## Author contributions

L.K.M. prepared the draft manuscript text and figures, prepared samples for 14C analysis and performed data analysis. M.S.A. assisted with project design, arranged bore access, provided information on groundwater flow directions and potential DOC sources, and assisted in the preparation of Fig. 4. M.I.B. contributed to the initial study plan, assisted with FT-ICR MS analyses, processed and analysed FT-ICR MS data, and provided manuscript comments. M.S.A., H.R., D.M.O., K.M. and A.B guided the study design, field sampling and provided manuscript comments. P.O., H.R., D.M.O., M.S.A. and L.K.M. undertook field sampling. P.O. undertook LC-OCD and fluorescence analyses, engaged in project planning, and provided information on adsorption and biodegradation of groundwater dissolved organic carbon at each site. A.M.M. undertook FT-ICR MS analyses and provided manuscript comments. R.G.M.S. provided input on the interpretation of FT-ICR MS data and provided manuscript comments. I.R.S and C.E.M. provided comments on the manuscript.

## Competing interests

The authors declare no competing interests.
