## [Peer Review File · Nature Communications]

REVIEWERS' COMMENTS

Reviewer #1 (Remarks to the Author):

Dear Authors,

I find the revisions satisfactory. I still rate the access to molecular oxygen (and attendant reactive oxygen species) as a critical decisive aspect of molecular evolution of DOM molecules. If absent, reaction trajectories will follow the course described here and in the previous GCA paper, because these have different kinetics but are not truly different.

Response to Reviewer Comments

Below is the final reviewer comment on the manuscript, reproduced verbatim, and our response. Please note that the line numbers refer to line numbers as shown in the track changed version of the manuscript.

Comments from Reviewer #1:

Dear Authors,

I find the revisions satisfactory. I still rate the access to molecular oxygen (and attendant reactive oxygen species) as a critical decisive aspect of molecular evolution of DOM molecules. If absent, reaction trajectories will follow the course described here and in the previous GCA paper, because these have different kinetics but are not truly different.

Author's response:

This comment refers to the ability for reactive oxygen species (ROS) to potentially also be produced non-photochemically in dark environments that are exposed to molecular oxygen or ROS, noting that in groundwater there is no physical mechanism of oxygen replenishment, so only the modest amount of oxygen originally dissolved are available along a particular flow path. We have added the following changes (shown in red) to the manuscript:

Line 83 – 91: *“Groundwater is often more reduced than other aquatic environments due to **biotic factors such as the progressive microbial consumption of molecular oxygen and other terminal electron acceptors (e.g. dissolved oxygen), and abiotic factors limiting oxidation, including the exclusion of sunlight which produces reactive oxygen species that can oxidize DOM***⁵⁰*. Research has shown that hydroxyl radicals can also be produced non-photochemically in dark environments*⁵¹ *due to the transfer of electrons from redox-active hydroquinone moieties to O₂*⁵²*, or due to Fe-mediated reduction of O₂*^{53,54}*. In both cases molecular oxygen is required, thus dark and continuously reduced conditions would prevent these reaction pathways from occurring.”*

Line 294 – 299: *“Our results show that the current paradigm of aged, highly processed, apparently stable DOM occurring in the center of H/C versus O/C space may actually be constrained to well-mixed, oxic, open waters, and that the oldest DOM appears instead to occur in dark anoxic aquifers **where molecular oxygen and attendant reactive oxygen species are unavailable.** ~~where~~ In these environments, the most persistent formulae are biolabile with high H/C, or contain low O/C including aromatic formulae.”*

We have also added the following references:

50 Waggoner, D. C., Wozniak, A. S., Cory, R. M., & Hatcher, P. G. The role of reactive oxygen species in the degradation of lignin derived dissolved organic matter. *Geochimica et Cosmochimica Acta*, 208, 171-184, doi:<https://doi.org/10.1016/j.gca.2017.03.036> (2017).

- 51 Page, S. E., Kling, G. W., Sander, M., Harrold, K. H., Logan, J. R., McNeill, K., & Cory, R. M. Dark Formation of Hydroxyl Radical in Arctic Soil and Surface Waters. *Environmental Science & Technology*, 47(22), 12860-128670, doi: <https://doi.org/10.1021/es4033265> (2013).
- 52 Page, S. E., Sander, M., Arnold, W. A., & McNeill, K. Hydroxyl Radical Formation upon Oxidation of Reduced Humic Acids by Oxygen in the Dark. *Environmental Science & Technology*, 46(3), 1590-1597, doi:<https://doi.org/10.1021/es203836f> (2012).
- 53 Zepp, R. G., Faust, B. C., & Hoigne, J. Hydroxyl radical formation in aqueous reactions (pH 3-8) of iron(II) with hydrogen peroxide: the photo-Fenton reaction. *Environmental Science & Technology*, 26(2), 313-319, doi:<https://doi.org/10.1021/es00026a011> (1992).
- 54 Lipson, D. A., Jha, M., Raab, T. K., & Oechel, W. C. Reduction of iron (III) and humic substances plays a major role in anaerobic respiration in an Arctic peat soil. *Journal of Geophysical Research*, 115(G4). doi:<https://doi.org/10.1029/2009JG001147> (2010).

END OF REVIEWS